# Effects of a New Combination of Natural Extracts on Glaucoma-Related Retinal Degeneration

**DOI:** 10.3390/foods10081885

**Published:** 2021-08-15

**Authors:** Claudio Molinari, Sara Ruga, Mahitab Farghali, Rebecca Galla, Rosario Fernandez-Godino, Nausicaa Clemente, Francesca Uberti

**Affiliations:** 1Laboratory of Physiology, Department of Translational Medicine, University of Piemonte Orientale, Via Solaroli 17, 28100 Novara, Italy; claudio.molinari@med.uniupo.it (C.M.); sara.ruga@uniupo.it (S.R.); mahitab.farghali@uniupo.it (M.F.); rebecca.galla@uniupo.it (R.G.); 2Ocular Genomics Institute-Massachusetts Eye and Ear, Harvard Medical School, Boston, MA 02115, USA; Rosario_FernandezGodino@MEEI.HARVARD.EDU; 3Dipartimento di Scienze della Salute, Interdisciplinary Research Center of Autoimmune Diseases-IRCAD, Università del Piemonte Orientale, 28100 Novara, Italy; nausicaa.clemente@med.uniupo.it

**Keywords:** vitamin D3, gastrodin, oxidative stress, glaucoma induction, RGC cells

## Abstract

Background: Glaucoma is currently the leading cause of irreversible blindness; it is a neuropathy characterized by structural alterations of the optic nerve, leading to visual impairments. The aim of this work is to develop a new oral formulation able to counteract the early changes connected to glaucomatous degeneration. The composition is based on gastrodin and vitamin D3 combined with vitamin C, blackcurrant, and lycopene. Methods: Cells and tissues of the retina were used to study biological mechanisms involved in glaucoma, to slow down the progression of the disease. Experiments mimicking the conditions of glaucoma were carried out to examine the etiology of retinal degeneration. Results: Our results show a significant ability to restore glaucoma-induced damage, by counteracting ROS production and promoting cell survival by inhibiting apoptosis. These effects were confirmed by the intracellular mechanism that was activated following administration of the compound, either before or after the glaucoma induction. In particular, the main results were obtained as a preventive action of glaucoma, showing a beneficial action on all selected markers, both on cells and on eyecup preparations. It is therefore possible to hypothesize both the preventive and therapeutic use of this formulation, in the presence of risk factors, and due to its ability to inhibit the apoptotic cycle and to stimulate cell survival mechanisms, respectively. Conclusion: This formulation has exhibited an active role in the prevention or restoration of glaucoma damage for the first time.

## 1. Introduction

Glaucoma is the leading cause of irreversible blindness in the world, and it is estimated that in 2020, approximately 80 million people had glaucoma, of which 11.2 million had a bilateral form [1,2]. This condition causes selective damage to retinal ganglion cells (RGC) [3], resulting in oxidative stress, apoptosis, and neuroinflammation [4]. For this reason, glaucoma should be considered a neuropathy characterized by specific structural degenerative alterations of the optic nerve, leading to visual impairments [5]. Studies conducted in recent years have shown that glaucoma is a complex chronic-degenerative disease with a multifactorial etiology, in which oxidative damage is only the “primum movens” of a series of pathogenetic and mutually connected events [6], leading to the generation of uncontrollable reactive oxygen species (ROS). This causes oxidative stress, which plays a crucial role in the pathogenesis of age-related eye diseases, by activating nuclear factor kappa B (NF-kB), vascular endothelial growth factor (VEGF), and lipid peroxidation. These factors lead to the production of inflammatory cytokines, angiogenesis, protein and DNA damage, and apoptosis [7]. In addition, several other factors can induce glaucoma, such as aging, genetic predisposition, exogenous or endogenous environmental factors [6], increased glutamate levels [8], alteration in nitric oxide (NO) metabolism [9], vascular alterations [10], hypoxia [11], oxidative stress [12], inflammation [13], and apoptosis of glial cells [14]. In this scenario, the optic nerve damage occurs due to an increase in intraocular pressure (IOP), resulting from either an increase in the production of aqueous humor or a decrease in its outflow [15]. A large increase in IOP can potentially cause a vascular occlusion, resulting in ischemia of the optic disc and subsequent loss of vision [16]. A hypothetical strategy to prevent the degenerative cascade may be to act in the initial phase of the disease. This should make it possible to reduce the number of cases, preventing the effects of the disease and slowing down its course. Today, there are several treatments for glaucoma, including invasive procedures such as laser trabeculoplasty and incisional surgery. Unfortunately, these methods have side effects that affect the quality of life [17]. For this reason, they are used only when the disease is in a severe state. In addition, there are topical pharmacological applications, exerting an indirect neuroprotective action able to reduce IOP (such as α2-agonists, β-antagonists/blockers, prostaglandin analogs, carbonic anhydrase inhibitors, and cholinergic agents) [18]. Moreover, recent studies have investigated the specific role of anti-edema therapies, obtaining some useful therapeutic effects in pre-clinical models of stroke [19,20]. In this unclear context, innovative and regenerative strategies are being studied to explore a new glaucoma treatment that is capable of maintaining patients’ quality of life [21,22]. In fact, recent studies have demonstrated the importance of supplementation treatment aimed at neuroprotection in patients at risk of glaucoma or with overt disease. For example, herbal medicine offers active ingredients that are useful to slow down the progression of the disease, by acting in its early stages [23,24]. The present study, therefore, aims to evaluate the effect of a mixture of natural substances on in vitro mouse glial and retinal cells subjected to experimental induction of typical lesions of glaucoma, using N-methyl-D-aspartate (NMDA) and H2O2 [25,26,27]. These substances, as reported in the literature, are widely used as glaucoma inductors [28,29], and if combined with each other can reproduce two important aspects of retinal degeneration, high pressure and oxidative stress. After obtaining the typical conditions of glaucoma in vitro, the effectiveness of treatment with a mixture of vitamin D3, gastrodin, lycopene, vitamin C, and blackcurrant extract was evaluated. Even if considered individually, these substances have important effects on ocular function. In particular, it is known that vitamin C has a beneficial effect on the eye [30] because it is involved in the control of IOP [31], as reported by the evidence collected in patients with open-angle glaucoma [32], and in the prevention of cataracts [33]. A correlation between the serum level of vitamin D3 and glaucoma and/or intraocular pressure (IOP) has also been reported [34], since vitamin D3 acts both by counteracting oxidative stress and promoting vascularization [35]. Vitamin D3 has been shown to play a major role in reducing inflammation, modulating the immune response, and decreasing angiogenesis in the eye. In particular, vitamin D3 could be a protective factor for glaucoma, and its deficiency could explain glaucoma occurrence or severity in some patients [36]. However, the specific role of vitamin D3 in the pathophysiology of glaucoma remains to be fully elucidated. Lycopene is an important natural compound belonging to the family of carotenoids, which is able to prevent cell death by acting directly on oxidative stress, with the subsequent block of inflammation and angiogenesis [37]. In addition, low levels of lycopene are linked to various forms of glaucoma, such as normal tension glaucoma (NTG) and primary open-angle glaucoma (POAG), supporting the hypothesis of its usefulness in the treatment of glaucoma [38]. Blackcurrant contains anthocyanins, which have therapeutic potential against hypertension and other cardiovascular, neurodegenerative, and ocular diseases [39]; in particular, its oral administration represents a safe and promising supplement for patients with glaucoma, and particularly those with impaired vision [40]. Indeed, blackcurrant anthocyanins have shown an ability to improve certain visual functions [41], and their oral administration has been found to increase blood flow in the optic nerve head (ONH) and to modify plasma concentrations of endothelin-1 (associated with the pathogenesis of glaucoma) [39]. Finally, gastrodin, the active ingredient of Gastrodia Elata which is a member of the family of Orchidaceae, is traditionally used for the treatment of cerebrovascular damage for its high-performance effects [42,43]. This is based on the increase in the activity of antioxidant enzymes and the reduction of oxidative stress, which together with the inhibition of caspase leads to the attenuation of apoptosis mechanism [44]. Gastrodin can also decrease the production of IL-6 and TNF-α by significantly reducing the expression of iNOS, suggesting a remarkable anti-inflammatory potential of this compound [45]. Another risk factor for glaucoma includes microglial activation, which increases inflammation, and contributes to loss of retinal ganglion cells. This pathogenetic mechanism is inhibited by gastrodin [46]. The aim of this study was to understand the effectiveness of a combination based on vitamin D3 and gastrodin, to counteract the course of RGC degeneration. We also wanted to verify the hypothesis that the synergistic effect of this compound can be amplified by vitamin C, lycopene, and blackcurrant extract, thus decreasing oxidative stress, preventing the apoptotic cycle, and promoting cell survival of retinal cells.

## 2. Materials and Methods

### 2.1. Retinal Ganglion Cells (RGC)

RCG cells were extracted from 8 to 10-week-old mice C57BL/6JOlaHsd (n=108), following a method reported in the literature [26,47]. The animals were obtained directly from the enclosure of the University of Eastern Piedmont, selecting them from those who were not included in other experiments and not subjected to any treatment. This project received regular authorization with n. 04/2020-UT of 04/03/2020, pursuant to Legislative Decree 26/2014; issued by the General Directorate of Veterinary Health and Food-Ministry of Health (breeding plant: Department of Health Sciences located in via Solaroli, 17 Novara, obtained from the veterinary and animal welfare service of the Municipality of Novara, prot. num. 0087803/2017 of 13/12/2017). Briefly, the eyeballs were extracted from the mouse’s orbit with tweezers, including a part of the optic nerve [43,44]. The retina was conserved in saline solution with 3% penicillin/streptomycin (Sigma-Aldrich, St. Louis, MO, USA, CN P4333) in an iced box. The retina tissue deprived of the sclera was put into Dulbecco’s modified Eagle’s medium (DMEM, Sigma-Aldrich, CN D5671) and was gently mechanically dissolved and centrifuged at 900 rpm for 10 min. The pellet was resuspended in DMEM supplemented with 5% fetal calf serum (FCS, Sigma-Aldrich, CN F7524), 1% penicillin/streptomycin, and 2 mM glutamine (Sigma-Aldrich, CN 59202C). At 72 h post isolation, non-adherent cells were removed, and the culture medium changed. In the following days, the cells were expanded while maintaining their pigmentation, and after 1 week they reached 60% confluence and in two weeks about 80%. Half of the medium was replenished every 2–3 days [48]. The experiments were carried out immediately upon reaching 80% confluence. Cells were maintained in the incubator at 37 °C with 5% CO_2_ for two weeks before the experiments. Cells were plated in different ways to perform several experiments: 1 × 10^5^ cell/well plated in 96-well to study cell viability by MTT test and ROS production; in petri dishes until confluence (80%) to analyze the intracellular pathways by Western blot analysis. Before stimulation, the cells were maintained overnight with DMEM without red phenol, and supplemented with 2% FCS, 1% penicillin/streptomycin, and 2 mM glutamine in the incubator at 37 °C with 5% CO_2_ and 95% humidity.

### 2.2. Preparation of Neuroretina Explants (Eyecup Preparations)

Bovine eyecup preparation is a well-known in vitro eye tissue model that replicates the anterior chamber of the eye [49,50], due to the high similarities between bovine and human visual systems as well as their retinal structure [51]. Retinae were isolated from bovine eyes [28,52] obtained from the food industry and placed in physiological ice saline solution supplemented with 1% penicillin/streptomycin for 2 h at 4 °C. The retinal explants were maintained in phenol red free DMEM supplemented with 2% FCS, 3% penicillin/streptomycin, and 2 mM glutamine in 6-well plates prepared with sterile foil to prevent retinal adhesion to the incubator. The next day, the retinal explants were treated for 7 days with natural compounds described below.

### 2.3. Agent Preparations

Both cells and tissues were treated with vitamin D3 (V, 100 nM) [53], lycopene (L, 20 µM) [37], vitamin C (C, 200 μM) [30], and blackcurrant extract (R, 200 mM) [39] following the methods reported in the literature. Gastrodin (G), on the other hand, was previously tested in a dose-response study on RGC, with doses from 0.1 µM to 100 µM to verify the most effective concentration. The 50 µM dose was then used in retinal explants [44]. All of these agents were dissolved in FM-LipoMatrix^®^, a new technology based on a patented solvent (patent N°102017000036744 by noiVita srls, produced by Pro-Bio INTEGRA srl, Rovigo, Italy), and were tested alone and in combination (in this work the compound was referred to as VGLCR) following the hypothesis of a new potential formula to be used in humans. The effects of VGLCR were studied before or after experimental induction of glaucoma by means of N-methyl-D-aspartate (NMDA, 300 μM, Sigma-Aldrich, CN M3262) and H_2_O_2_ (2 mM, Sigma-Aldrich, CN H1009) on both cells and tissues. NMDA and H_2_O_2_ are important well known glaucoma inductors [28] capable of mimicking high pressure and oxidative damage, as occurs in humans.

### 2.4. Experimental Protocol

RGC cells and retinal explants were used to study different biological aspects involved in glaucoma in three phases. In the first one, cells and tissue were treated with agents (vitamin D3, gastrodin, lycopene, vitamin C, and blackcurrant) alone and in combination (VGLCR) for 8 days, to evaluate the effectiveness of the agents under physiological conditions [54] and analyze cell viability by MTT test. In addition, cell viability on RGC cells was analyzed periodically every 2 days for 10 days, to verify the correct timing of stimulation. In a second set of experiments, RGC cells and retinal explants were pre-treated with NMDA (300 μM) and H_2_O_2_ (200 µM) for 4 days [28,29] alone and together, and then stimulated with all compounds alone and combined for 8 days, to mimic high pressure and oxidative damage, as occurs in humans [31]. In the same phase, inductors were also added after 8 days, and in the presence of VGLCR for a further 4 days as a post-stimulation, to study a possible protective mechanism exerted by the combination by analyzing cell viability and ROS production. In addition, the main intracellular pathways involved in neuroinflammation and oxidative stress were investigated. In this model, exposure to inductors led to negative effects on retinal cells that were capable of simulating oxidative stress-related cellular degeneration, similar to that which occurs in glaucoma, resulting in a realistic view of degenerative processes for 8 days [31]. Finally, the last series of experiments was carried out to verify the effectiveness of the combination (VGLCR) directly on retinal explants before and after the in vitro induction of glaucoma, by analyzing the effects at the microscopic level and the main intracellular pathways involved in the early stages of the disease.

### 2.5. MTT Test

MTT-based In Vitro Toxicology Assay Kit (Sigma-Aldrich, CN TOX1-1kt) was performed on a 96 well-plate, as reported in the literature [55]. Briefly, after stimulations, the cells were incubated with 3 mg/mL of MTT dye in DMEM white for 2 h at 37 °C in an incubator, and then purple formazan crystals were dissolved in equal volume of MTT solubilization solution. Cell viability was determined by measuring the absorbance at 570 nm with correction at 690 nm, through a spectrometer (VICTORX4 Multilabel Plate Reader, PerkinElmer, Waltham, MA, USA), and calculated by comparing results to control cells (baseline 0%).

### 2.6. ROS Production

The rate of superoxide anion production was determined as a superoxide dismutase-inhibitable reduction of cytochrome C, following a standard technique [55]. In all samples (treated and untreated), 100 µL of cytochrome C (Sigma-Aldrich, CN C3131) were added, and in another sample, 100 μL superoxide dismutase (Sigma-Aldrich, CN S9697) were also added for 30 min while maintaining in the incubator. The absorbance was measured at 550 nm using a spectrometer (VICTORX4, Multilabel Plate Reader), and O_2_ was expressed as mean ± SD (%) of nanomoles per reduced cytochrome C per microgram of protein, compared to the control (0 line) and expressed as percentage (%).

### 2.7. ERK Activation Assay

ERK/MAPK activity was measured by the InstantOne™ ELISA (Thermo Fisher, CN 85-86195-11) on cell lysates, following the manufacturer’s instructions [55]. Briefly, at the end of stimulations, cells were lysed with 100 μL Cell Lysis Buffer Mix, 50 μL/well of each sample was tested in InstantOne ELISA microplate strips, and at each well the antibody cocktail was added and incubated for 1 h at room temperature on a microplate shaker. At the end, the detection reagent was added to each well, and after 20 min, the reaction was stopped by adding stop solution to each. The strips were measured by a spectrometer (VICTOR X4, Multilabel Plate Reader) at 450 nm. The results were expressed as mean absorbance (%) compared to the control (0 line).

### 2.8. p53 Activity

At the end of stimulations, p53 activity was measured by p53 transcription factor assay kit (Cayman Chemical, CN 600020), directly on nuclear extracts obtained following the manufacturer’s instructions, as reported in the literature [55]. Briefly, cells were lysed with ice-cold 1X Complete Hypotonic Buffer, supplemented with NP-40 and centrifuged at 12,000 g at 4 °C for 10 min. The pellet was solubilized with ice-cold Complete Nuclear Extraction Buffer 1×, supplemented with protease and phosphatase inhibitors and then centrifuged at 12,000 g for 15 min at 4 °C; the supernatant was examined to analyze the activity of p53 related to the protein quantification through the BCA assay (Thermo Fisher. Whaltam, MA, USA, CN 23225).

### 2.9. MMP-9 Quantification Assay

MMP-9 ELISA Kit (DBA, Italy, CN KA0398) was used to quantify MMP-9 on culture supernatant samples, following the manufacturer’s instructions [56]. Briefly, samples in 96-well/plate were incubated at 37 °C for 90 min, and then biotinylated antibody and ABC working solution were added and the plate was incubated at 37 °C for 60 and 30 min, respectively. TMB substrate and stop solution were added after incubation for 15 min at 37 °C. The MMP-9 concentration was determined by measuring the absorbance through a spectrometer (VICTOR X4 Multilabel Plate Reader) at 450 nm, and calculated by comparing results to MMP-9 standard curve.

### 2.10. p38 Quantification Assay

The p38MAPK (total) and p38MAPK (Thr(P)-180/Tyr(P)-182) ELISAs kit (Biosource, CN KHO0061 and 85-86022-11) were performed as suggested by the manufacturer’s instructions. Briefly, after stimulations cells were lysed with a specific buffer kit [57] and a 1:25 dilution of the lysate was applied to the p38MAPK 96-well plate coated with primary antibody, along with standards and controls. Phosphorylated or total p38MAPK was detected by using the supplied detection antibody, horseradish peroxidase-conjugated as a secondary antibody, and stabilized chromagen. Absorbance was measured at 450 nm using VICTOR X4 Multilabel Plate Reader. The amount of phosphorylated and total p38MAPK was determined by reading the measured absorbance from a standard curve. p38MAPK (total) was measured by ELISA to control for the amount of protein assayed between samples, and the amount of phosphorylated p38MAPK (Thr(P)-180/Tyr(P)-182) was adjusted accordingly.

### 2.11. Detection of JNK Activity

JNK activity was measured in RGC cells and retinal tissue homogenates, using the Abcam JNK1/2 (pT183/Y185) +Total JNK1/2 ELISA kit (abcam CN 176662) following the manufacturer’s instructions. Briefly, 50 µL of sample was added to the wells and incubated for 1 h with 50 µL of Antibody Cocktail. After washing, 100 µL of TMB substrate was added to each and incubated for 15 min; at the end of this time, 100 µL of stop solution were used to detect JNK activity at 450 nm using VICTOR X4 Multilabel Plate Reader.

### 2.12. SOD Assay

The SOD assay measures all three types of SOD (Cu/Zn, Mn, and FeSOD), as reported by manufacturer’s instructions (Cayman’s Superoxide Dismutase Assay Kit, CN 706002). In a 96-well, the level of SOD was measured comparing data obtained from tissue lysates to a standard curve (0.05–0.005 U/mL). The absorbance of all samples was measured through a spectrometer (VICTOR X4 Multilabel Plate Reader) at 480 nm, and the results were expressed as means (%) compared to control [58].

### 2.13. Quantification of Retinal Layer

The pictures of the retinal explants at 0 and 8 days were taken using Nikon D70 camera, and the areas were quantified using ImageJ program [28]. The retinal areas after each stimulation were compared to day 0 (T0, area 100%) and to the untreated sample (control). The results are expressed as a percentage of the relative reduction area, which was compared to the original size, similar to that used for the wound healing area measurement in a previous paper, using the following formula [56,59]:% area = [(WA0 − WA)/WA0] × 100(1)
WA0 = original size of retinal explanted(2)
WA = area after each stimulation(3)

### 2.14. Western Blot

After stimulation, RGC cells were lysed on ice with Complete tablet buffer (Roche, CN 11836145001), supplemented with 2 mM sodium orthovanadate (Na3VO4, Sigma-Aldrich, Milan, Italy, CN S6508), 0.1 M sodium fluoride (Sigma-Aldrich, Milan, Italy, CN 450022), 1:1000 phenylmethanesulfonyl fluoride (PMSF, Sigma-Aldrich, Milan, Italy, CN P7626), and 1:100 mix protease inhibitor cocktail (Sigma-Aldrich, Milan, Italy CN S8820). In total, 35 µg of proteins from each lysate were resolved into 10% SDS-PAGE gels, and transferred to polyvinylidene fluoride membranes (PVDF, GE Healthcare Europe GmbH, Milan, Italy) which were incubated overnight at 4 °C with a specific primary antibody: anti-Annexin V (1:1000 Santa Cruz, CN sc-74438), anti-iNOS (1:1000 Santa Cruz, CN sc-7271), and anti-Bax (1:250, Santa Cruz, CN sc-7480). Retinal explants were plated in a 60 mm dish, and after stimulation were lysed in ice with 100 mg tissue/150 μL of lysis buffer (0.1 M Tris, 0.01 M NaCl, 0.025 M EDTA, 1% NP40, 1% Triton X100, Sigma-Aldrich, Milan), supplemented with 2 mM sodium orthovanadate (Na3VO4, Sigma-Aldrich, Milan, Italy, CN S6508), 0.1 M sodium fluoride (Sigma-Aldrich, Milan, CN 450022), 1:100 mix of protease inhibitors (Sigma-Aldrich, Milan, CN S8820), 1:1000 phenylmethylsulfonyl fluoride (PMSF; Sigma- Aldrich, Milan, CN P7626), using an electric potter at 1600 rpm for 2 min. Samples were mixed for 30 min at 4 °C, centrifuged for 30 min at 13000 rpm at 4 °C, and 40 µg from each tissue lysates were resolved into 10% SDS-PAGE gels, and transferred on PVDF membranes which were incubated overnight at 4 °C with a specific primary antibody: anti-Bax (1:250, Santa Cruz, CN sc-7480), anti-iNOS (1:1000 Santa Cruz, CN sc-7271), anti-SIRT1 (1:1000, Sigma-Aldrich, CN SAB4301426), and anti-OPA1 (1:250, Santa Cruz, CN sc-393296). Protein expressions of both RGC and retinal lysates were normalized and verified through anti-β-actin detection (1:5000, Sigma-Aldrich, CN A5441). The results are expressed as a mean ± SD (% vs. control).

### 2.15. Statistical Analysis

The data were processed using Prism GraphPad statistical software for normalization, peak picking, and for group comparison. The images were produced directly by BFB-Z and ImageJ. All results were analyzed by one-way analysis of variance (ANOVA), followed by Bonferroni post hoc test and all results were expressed as mean ± SD of at least 4 independent experiments produced in triplicates. Comparisons between the two groups were performed using a two-tailed Student’s t-test. Multiple comparisons among groups were analyzed by two-way ANOVA followed by a two-sided Dunnett post hoc test. Differences were considered statistically significant with a *p* value < 0.05.

## 3. Results

### 3.1. Analysis of Cell Viability and ROS on RGC Cells Treated for 8 Days

The MTT test was performed in a dose-response study to identify the best concentration of gastrodin (G) to be used in all subsequent experiments on RGC cells. A concentration in the range of 1 to 100 µM was chosen for 8 days. As reported in Figure 1A, G was able to induce a significant increase (*p* < 0.05) in cell viability compared to the control. In particular, 50 µM of this substance induced the best significant increase (*p* < 0.05), compared to the other concentrations tested (about 18.9% with 1 µM; about 24.2% with 10 µM; and about 12.3% with 100 µM). For this reason, G 50 µM was maintained for all subsequent experiments. Additional experiments were carried out to exclude any cytotoxic effect induced by vitamin D3 (V, 100 nM), gastrodin (G, 50 μM), lycopene (L, 15 µM), vitamin C (C, 200 μM), and blackcurrant (R, 200 µM) alone and together, analyzing cell viability after 8 days of stimulation on RGC cells. As shown in Figure 1B, all these substances tested alone were able to induce a significant increase in mitochondrial metabolism (*p* < 0.05) compared to the control. This provides the scientific basis for hypothesizing a new formulation, based on the combination of these substances. In addition, the combination of V 100 nM, G 50 μM, L 15 µM, C 200 μM, and R 200 µM, named VGLCR, confirmed this hypothesis. After 8 days of treatment, VGLCR was able to amplify (*p* < 0.05) the beneficial effect observed using single compounds. In particular, VGLCR had effects greater than about 50% compared to R and G alone, about three times compared to C, about four times compared to L, and three times compared to V. Moreover, further experiments were carried out to explore the effects of the substances either alone or combined together over 10 days, to verify the correct time point of stimulation as reported in Appendix A. In particular, the results on cell viability confirmed that the greatest effects were obtained after 8 days of stimulation for all treatments, and VGLCR confirmed to have a better action throughout the analyzed period compared to all the other substances (*p* < 0.05). These data confirmed the absence of adverse effects during the treatment and the effectiveness of the combination, suggesting a synergistic effect exerted by the sum of all compounds. Furthermore, since reactive oxygen species (ROS) have been implicated in the pathogenesis of various eye diseases, including mitochondrial dysfunction, ROS production after 8 days of stimulation with all compounds alone or combined was investigated. As shown in Figure 1C, the antioxidant property of V 100 nM, G 50 μM, L 15 µM, C 200 μM, and R 200 µM alone was confirmed, and VGLCR produced an amplified effect (*p* < 0.05) with respect to individual agents and control. All these data confirm the effectiveness of VGLCR on mitochondrial balance, supporting the hypothesis that it can be used to prevent retinal cell loss. For this reason, only the VGLCR combination was tested in all subsequent experiments.

### 3.2. Cell Viability and ROS Production on RGC Cells Subjected to Conditions Mimicking Glaucoma

Since VGLCR may be used both before and during the treatment of glaucoma, some experiments on RGC cells were performed before and after the induction of cell damage, mimicking glaucoma by means of NMDA and H_2_O_2_. These agents are classical glaucoma inductors capable of mimicking the damage caused by high ocular pressure and oxidative damage, and for this reason they have been added alone or together for 8 days. As shown in Figure 2A, NMDA and H_2_O_2_ alone confirmed their role as glaucoma inductors, reducing cell viability (*p* < 0.05 vs. control) and indicating the presence of an injury to RGC cells. The simultaneous treatment with NMDA and H_2_O_2_ after 8 days amplified this reduction in cell viability (*p* < 0.05) compared to NMDA and H_2_O_2_ alone (about 69% and 88%, respectively). This stimulation indicates a severe damage of RGC cells, similar to the human glaucomatous condition. To demonstrate the effectiveness of VGLCR as a therapeutic adjuvant, both during the early stage of this condition and during the overt disease the stimulation was performed before and after NMDA and H_2_O_2_ alone and together. In all conditions tested, VGLCR was able to restore the induced damage, confirming a positive role on cell viability compared to control and glaucoma inductors (*p* < 0.05). In particular, the main effects were observed when NMDA and H_2_O_2_ were added together, proving the existence of a beneficial effect of VGLCR against a glaucomatous condition. At the same time, VGLCR appears to have more of an effect (*p* < 0.05) when added prior to glaucoma induction, using NMDA and H_2_O_2_ together, indicating a possible role in the early stage of the disease. Similarly, the analysis of ROS production, shown in Figure 2B, confirmed the negative effects of NMDA and H_2_O_2_ alone and together (*p* < 0.05 vs. control). In particular, significant damage (*p* < 0.05) was obtained by summing together the glaucoma inductors. VGLCR was able to prevent and restore all these negative conditions, confirming what was observed in cell viability. Since the main negative effect was obtained with the administration of NMDA+ H_2_O_2_, only this treatment was used to induce the damage on all subsequent experiments.

### 3.3. Analysis of the Main Intracellular Pathways Activated on RGC Subjected to Conditions Mimicking Glaucoma

Since previous findings have demonstrated the ability of VGLCR to induce beneficial effects on retinal cells, it was important to verify the intracellular mechanisms leading to these effects. This involved analyzing the activation in vitro of main pathways linked to glaucoma. As reported in Figure 3, the effects of VGLCR before and after the administration of NMDA + H_2_O_2_ on the activity of p53 and on the expression of Annexin V and Bax was investigated. The summed glaucoma inductors confirmed their negative effects on RGC cells, and the subsequent cell loss is reported in Figure 3A for p53 activity compared to the control (*p* < 0.05). Conversely, VGLCR alone did not activate p53 (*p* > 0.05 vs. control), but if added before or after the glaucoma inducers it was able to counteract the adverse effects (*p* < 0.05 vs. NMDA+ H_2_O_2_, approximately eight times and seven times greater in both conditions, respectively). In particular, the combination appears to be able to induce a greater effect if added before rather than after the induction of damage (about 30%), but this difference is not statistically significant, revealing a limitation of these in vitro experiments. In addition, this beneficial effect was also confirmed by analysis of Annexin V and Bax, as reported in Figure 3B,C,E. NMDA + H_2_O_2_ induced a significant increase in Annexin V expression compared to the control (*p* < 0.05), supporting data on the nuclear compartment damage of RGC cells that can cause cell death, as suggested by the observations in cell viability, ROS analysis, and p53 activity. Conversely, VGLCR added before or after the injury was able to reduce this expression by approximately four times and approximately three times, respectively (*p* < 0.05), indicating its protective effects on RGC cells. The effects obtained before or after the administration of VGLCR support the hypothesis that this combination has a greater effect if added before the damage (about 38% compared to after), but these data need to be confirmed in a more complex experimental system. Furthermore, the analysis of Bax (Figure 3C,E) shows a significant increase in its expression (*p* < 0.05) compared to the control in the presence of glaucoma inductors, indicating the presence of RGC cell apoptosis and confirming that the association of NMDA + H_2_O_2_ mimicked the real negative consequences of glaucoma in humans. VGLCR was able to counteract these negative effects if added both before and after the injury, reducing (*p* < 0.05) the expression of Bax compared to that which happens after the inductors (about five times if added before and four times if added after, respectively). Among the effects obtained by adding VGLCR either before or after, the results confirmed the hypothesis that it has a great effect if added before the damage (about 27% compared to after), but this data need to be confirmed. These findings demonstrated for the first time that NMDA and H_2_O_2_ combined reproduced an in vitro glaucomatous condition, and that VGLCR was able to prevent RGC cell loss.

At this point, since oxidative stress is involved in cell apoptosis, an analysis of iNOS was performed to confirm the role of its modulation in preventing cell death. As illustrated in Figure 3D,E, NMDA + H_2_O_2_ induced a significant increase in iNOS expression (*p* < 0.05) compared to the control, supporting previous data on ROS production, p53 activity, and both Annexin V and Bax expressions. Conversely, stimulation with VGLCR before and after damage was able to reverse this negative condition (about five times if added before, and three times if added after). This supports the hypothesis that the antioxidant activity is maintained by VGLCR, and this counteracts oxidative stress. In particular, the best effect appears to occur when VGLCR is administered before damage (approximately 81% compared to after), but this needs to be confirmed. Among the negative consequences, the alteration of the activity of MMPs seems to be very important, leading to an altered composition of the extracellular matrix, which involves MMP9-activity (Figure 3F). NMDA + H_2_O_2_ confirmed the ability to induce the damage by activating MPP9 (*p* < 0.05 vs. control), leading to cells loss; VGLCR was able to significantly decrease the MPP9 activation when added both before (about 4.4 times compared to damage) and after the injury (about 4 times compared to damage), confirming its ability to improve cell viability by modulating oxidative stress and apoptosis of RGC cells. Pre-treatment with VGLCR confirmed an approximately 19% better effect than post-stimulation, but further experiments may be needed to fully clarify this observation.

All these findings demonstrated for the first time that VGLCR is able to reverse cell degeneration and prevent cell death, by modulating the intracellular pathways involved in human glaucoma.

Some additional experiments were carried out to evaluate the activity of MAPK/kinases. In particular, ERKs/MAPK, p38 MAPK, and JNK1/2 activities were studied, as they are commonly involved in the early stage of optic nerve degeneration. As reported in Figure 4A, the activation of cell survival kinase (ERKs/MAPK) was significantly decreased in the presence of glaucoma inductors, confirming the previously observed negative consequence of the stimulation with NMDA + H_2_O_2_ (*p* < 0.05 vs. control). On the contrary, the presence of VGLCR, administered both before and after the injury, was able to significantly increase this activity, supporting the survival response of RGC cells (*p* < 0.05 compared to inductors). Furthermore, the effects on p38/MAPK (Figure 4B), a stress kinase, showed similar effects; NMDA+H_2_O_2_ significantly increased the activity of this cellular stress marker compared to the control (*p* < 0.05), and the presence of VGLCR before and after the damage reverted this negative effect, bringing the activity back to the control values. This confirmed that the inhibition of p38/MAPK activity in the retinal field may represent a strategical therapeutic target for preventing the early stage of pathogenesis in optic neuropathies (*p* < 0.05 vs. NMDA + H_2_O_2_). Finally, the analysis of JNK1/2 activity (Figure 4C) showed that NMDA + H_2_O_2_ promoted the degenerative processes by increasing JNK activity (*p* < 0.05 vs. control). On the contrary, the presence of VGLCR before or after the damage reduced this activation, leading to control values and supporting its ability to prevent and/or restore the injury caused by glaucoma inductors (*p* < 0.05 vs. NMDA+ H_2_O_2_).

Some additional experiments were performed to verify the effectiveness of VGLCR directly administered on complex systems as retinal tissues explants (Figure 5). In these experiments of an in vitro experimental model of retinal tissue, the same treatments and conditions used on RGC cells were maintained, analyzing the effects of NMDA + H_2_O_2_ alone and in the presence of VGLCR added either before or after the experimental injury. The macroscopic analysis of the eyecup preparations and the relative measurement of the areas showed that the glaucoma inductors were able to degenerate retinal tissue, thus leading to cell death (*p* < 0.05 compared to control). This observation confirmed the data obtained on RGC cells. In addition, several morphological changes in the tissues treated with VGLCR were observed compared to the control and to what was observed after administration of NMDA + H_2_O_2_. Indeed, VGLCR administered before the induction of the glaucomatous lesion was able to prevent the retinal degeneration (*p* < 0.05 compared to glaucoma inductors) better than when the compound was added after the damage (about 35%). At morphological level, in retinal tissues treated with VGLCR, the ophthalmic artery was always visible both with respect to the control and after the stimulation with glaucoma inductors, confirming the hypothesis that VGLCR was able to counteract and/or prevent the negative aspects of glaucoma, making it suitable for supplementing glaucoma therapy.

### 3.4. Beneficial Role of VGLCR to Prevent/Restore the Damage Caused by NMDA + H_2_O_2_ in Retinal Extracts

Further experiments were carried out directly on the retinal tissue treated as previously described, in order to analyze the effects of VGLCR on the main kinases involved in retinal degeneration and typical of human glaucoma. As reported in Figure 6A, the level of p53 activity confirmed the loss of retinal integrity, indicating a consequence of administration of NMDA + H_2_O_2_. VGLCR was able to revert degradation of this tissue by reducing the activation of p53 when added before and after NMDA+ H_2_O_2_ (*p* < 0.05; by about nine times and five times, respectively), confirming its beneficial effects. Analysis of data collected before and after the induction of damage suggested that pre-treatment was better than post-stimulation (approximately 75%, *p* < 0.05), supporting the hypothesis of a potential use as a supplement with preventive action in humans. Since oxidative stress is considered as a *primum movens* of glaucomatous degeneration, the most important molecular pathways implicated in this critical condition, such as SOD activity (Figure 6B) and iNOS (Figure 7A,E), were also analyzed. Data obtained demonstrated that the SOD level is higher in the presence of NMDA + H_2_O_2_ (*p* < 0.05 compared to control), and this negative effect was counteracted by the pre-treatment with VGLCR (*p* < 0.05, about 4.5 times), but also significantly reduced by the post-stimulation with the same combination (*p* < 0.05, about five times). Similarly, the analysis of iNOS expression confirmed the negative effects of oxidative stress on retinal tissues treated with NMDA + H_2_O_2_ (*p* < 0.05 vs. control), and the ability of VGLCR to prevent (*p* < 0.05, about 5.5 times) or restore the damage (*p* < 0.05, about four times). The greater protective action of VGLCR was again confirmed in these experiments; indeed, the pre-treatment induced a greater reduction than the post-stimulation (about 46%, *p* < 0.05). Starting from tissue microscopy observations, in which retinal remodeling and tissue death were observed, additional experiments to investigate MMP9 activity (Figure 6C) and Bax expression (Figure 7B,E) were performed. NMDA+H_2_O_2_ confirmed the degeneration and loss of retinal tissue as reported above (*p* < 0.05 vs. control), and these effects were significantly reduced by the stimulation with VGLCR before (p<0.05 about five times and about four times, respectively) and after the injury (*p* < 0.05 about four times and three times). In addition, the pre-treatment with VGLCR exerted a better influence than the post-stimulation (*p* < 0.05 about 26% and 40%, respectively), confirming the findings obtained from experiments on RGC cells and from observations at the macroscopic level. Furthermore, the MAPK pathway implicated in the early stage of glaucoma has been tested on eyecup preparations. As reported in Figure 6D,E, the p38MAPK and JNK1/2 activities were significantly increased by the presence of NMDA + H_2_O_2_ (*p* < 0.05 compared to control), confirming the negative consequences of oxidative stress on retinal tissues. On the other hand, VGLCR added both before and after the damage was able to prevent (*p* < 0.05, about two times and three times, respectively) and/or restore (*p* < 0.05 about 2 times and about 1.5 times, respectively) the glaucomatous condition, confirming its beneficial role to modulate oxidative stress and relative damage at the early stage of glaucoma. Again, in these cases, the main positive modulations on both markers were obtained by adding VGLCR before the induction damage (approximately 17% and 90%, respectively) rather than by adding it after injury. Furthermore, since NMDA + H_2_O_2_ could participate in the development of glaucoma through the inhibition of ERK activity (*p* < 0.05 vs. control; Figure 6F), the observation that this was prevented by pre-treatment with VGLCR (*p* < 0.05) or restored by the post-stimulation with the same combination (*p* < 0.05) is an important observation. Finally, it must be emphasized that the pre-treatment with VGLCR can slow down the adverse effects caused by glaucoma inductors better than the post-stimulation (*p* < 0.05 about 70%), and this supports its potential role in counteracting glaucoma.

### 3.5. Ocular Damage Analysis

To determine the involvement of SIRT1, a marker of ocular aging, and OPA1, a marker of ocular atrophy, Western blot on tissue lysates obtained from eyecup preparations were performed. As shown in Figure 7C,E, NMDA + H_2_O_2_ induced a significant decrease in SIRT1 phosphorylation compared to the control (*p* < 0.05), and the pre-treatment with VGLCR was able to reduce this negative effect better than the post-stimulation (*p* < 0.05, about 17%). These data define a novel role for SIRT1 as an important regulator of inflammation or oxidative response under a glaucoma condition. Finally, the glaucoma inductors were able to degrade retinal tissue (*p* < 0.05), but VGLCR pre-treatment was able to counteract this process (*p* < 0.05) by overexpressing OPA1 better than the post-stimulation (*p* < 0.05, about 50%), confirming its protective role (Figure 7D,E). All these findings support the hypothesis that VGLCR can protect the optic nerve, and, consequently, retinal cells by preventing ocular damage. Furthermore, VGLCR was also able to induce a physiological mechanism that allows the slowing of degeneration, as demonstrated by molecular mechanisms.

## 4. Discussion

Glaucoma includes a group of eye disorders that can lead to progressive and irreversible blindness [60]. Glaucoma damage is generally caused by an increase in intraocular pressure, which is supported by other factors, such as oxidative stress, leading to a progressive degeneration of the retinal ganglion cells and the optic nerve [6]. Oxidative stress plays a crucial role in the pathogenesis of age-related eye diseases, leading to the production of inflammatory cytokines, angiogenesis, proteins and DNA damage, and, ultimately, apoptosis. Dietary supplementation of plant natural products has demonstrated preventive and therapeutic effects, based on their capacity to scavenge free radicals and reduce enzymes involved in ROS production. It neutralizes the oxidative reaction that occurs in photoreceptor cells and upregulates the antioxidant defense system. In particular, some extracts of herbal origin have also shown their capacity to reduce the opacification of the lens and the apoptosis of retinal cells, and are able to inhibit the inflammatory markers of the blood-retinal barrier and improve ocular blood flow [7]. Deficiencies of specific nutrients have been found in patients with glaucoma, and dietary supplementation may play an important role in treatment [61]. Indeed, the goal of any strategy to prevent or modulate the degenerative cascade could be to act in the initial phase of the disease. Based on this, innovative strategies are further intensively investigated, to explore an innovative treatment of glaucoma that is capable of reducing the loss of functional retinal pigment cells in order to maintain patients’ quality of life [21,22,62,63]. Recent studies show the importance of integrative therapy in patients at risk or with glaucoma [23,24]. Mimicking human diseases in animal models in species, such as mice and bovines, has been a key resource to understand the pathogenesis of human eye diseases, and to develop novel therapeutic and drug delivery strategies [64]. Recently, the use of in vitro models for the study of glaucoma has opened up the possibility of studying the cellular and molecular mechanisms, which can help clarify the onset and progression of this disease [65].

In particular, the main models used in this research field include a wide variety of cell cultures, from cell lines to more complex models, such as tissue cultures (retinal organotypic cultures) and ex vivo preparations (in vitro preparations of ocular tissues) [65]. Furthermore, recent advances in regenerative medicine have led to the generation of 3D organic tissues (organoids) as organ-like structures to simulate a complex biological system. The development of retinal organoids is highly promising in regenerative medicine; organoids can be expanded and differentiated in vitro from ESCs and iPSCs [66,67,68]. If comprehensive, safe, and efficient protocols for the handling of organoids are yet to be developed, RGC cells have confirmed their key role in the glaucoma condition [65]. In this context, the efficacy of new nutraceutical compounds was tested on both in vitro (RGC cell) and ex vivo models (eyecup preparation). The combination of vitamin D3, gastrodin, lycopene, vitamin C, and blackcurrant (VGLCR) exerts a synergistic effect on retinal ganglion cells, indicating a possible new strategy to act in the early stage of glaucoma. Furthermore, reactive oxygen species have been implicated in the pathogenesis of various eye diseases, including mitochondrial dysfunction, which plays a crucial role in RGC cell degeneration; in this context, VGLCR is able to work on mitochondrial balance, supporting the hypothesis that this combination can be used to prevent cell loss. Since the main theory behind glaucoma concerns oxidative stress and IOP, further experiments were carried out in experimental models mimicking glaucomatous conditions by NMDA and H_2_O_2_. The pre- and post-treatment with VGLCR improved cell viability and reduced ROS production, thanks to the beneficial effects of VGLCR with a direct action on the activity of retinal cells by inhibiting cell loss. The data collected show that VGLCR is able to restore glaucoma-related damage caused by specific inductors (such as NMDA and H_2_O_2_) by promoting cell survival. Furthermore, the specific intracellular pathway activated by VGLCR during the glaucoma condition was analyzed, starting from the activity of p53. Our results show that there is a reduction in p53 activity following VGLCR treatment, and this effect is related to the reduction of nuclear compartment damage and apoptosis. Furthermore, since oxidative stress is notoriously involved in cell apoptosis, iNOS analysis was performed to confirm that its modulation can prevent cell death. The pre- and post-administration of VGLCR improved cell survival, by reducing iNOS expression caused by the glaucoma inductors. In addition, the analysis of MMP9 demonstrated the ability of VGLCR to restore the alteration of the composition of the extracellular matrix caused by glaucoma, both in RGC and in eyecup preparations. According to the literature, structural changes are believed to anticipate functional loss, involving alterations in transcription factors and extracellular signaling pathways, and inflammatory cytokines. NDMA-induced retinal toxicity involves both JNK and p38 MAPK, with the inhibitors of each being found as protective in this study. Similarly, JNK inhibition was protective against RGC loss in another ocular hypertensive model in a dose-dependent manner [69]. As reported above, the presence of VGLCR is able to significantly activate p38/MAPK and JNK, favoring the survival of RGC cells and eye tissues when added both before and after injury. This demonstrates its ability to prevent and/or restore the damage induced by glaucoma and confirms that inhibition of p38 MAPK signaling in the retina could represent a therapeutic target to prevent the early stages of the disease. In this context, some further experiments were performed to verify the effectiveness of VGLCR directly on complex systems, such as explants of retinal tissues. Since elevated IOP is a major risk factor, animal models relevant for glaucoma include culture of RGC and optic nerve experimental damage induced by ocular hypertension [65]. Indeed, many of the animal models of glaucoma presented an elevated IOP by reducing the outflow of the aqueous humor [70,71]. The opportunity to study whole tissue cultures has clear advantages compared to a monolayer of cultured cells, allowing the study of cell-to-cell interactions and offering the possibility to maintain an anatomical structure under glaucoma conditions [72]. Thus, to understand the pathological changes underlying this disease, we have set up an ex vivo model based on bovine retinas to analyze several mechanisms activated by NMDA and H_2_O_2_ [30,73]. Due to the high number of similarities between bovine and human visual systems as well as their retinal structure, these retinae appear to be a very promising alternative to animal experiments in ophthalmologic research [74]. Another important advantage of the use of bovine model is that in contrast to conventional cell culture models, the retina itself can be cultivated for some time. The culture of retinal tissue allows the maintenance of interactions and connections between neurons within the retina, and, for this reason, this model is very suitable for the screening of new therapeutic approaches [51]. Bovine eyecup preparation is a widely accepted model to reproduce the anterior chamber of the eye of glaucoma patients. This model is able to sustain a high pressure to assess some cellular responses [65]. Based on the advantages described above, we decided to also test the effects of VGLCR in this ex vivo model. In our experiments, it was possible to measure a significantly higher retinal mortality rate after NMDA and H_2_O_2_ treatment, while VGLCR administered both before and after obtaining the glaucoma injury was able to prevent retinal degradation. The beneficial role of this combination was also confirmed by the reduction of p53, exerted by the administration of VGLCR. Useful strategies could be to enhance the production of antioxidant enzymes, reduce ROS, or promote cytoprotective signaling pathways. With its antioxidant properties, this formulation could be used as a complementary therapy with a preventive effect on ROS production [75]. Different studies provide cumulating evidence, which supports the association of ROS with different aspects of the neurodegenerative process [76]. RGC cells are known to exhibit unique characteristics for their antioxidant defense mechanisms [77]. However, decreased ROS generation further promotes the survival of these cells [76]. Although currently available glaucoma therapy focuses on reducing IOP, some patients do not respond to this type of treatment, and research into RGC neuroprotection is emerging as a new therapeutic strategy. One of these strategies is precisely the reduction of oxidative stress [78]. Therefore, VGLCR demonstrates that it can restore glaucoma-induced oxidative damage by reducing the activity of iNOS and SOD. Furthermore, even if the presence of glaucoma caused damage to the retinal tissue, the response to the pre- and post-treatment with VGLCR can significantly activate the cell survival mechanisms in the early phase of the disease, by reducing the activation of p38, MAPK, and JNK. Indeed, these effects were mediated by the inhibitory effect induced by VGLCR on ocular oxidation (SIRT1) and ocular damage (OPA1).

## 5. Conclusions

In conclusion, all these tests have different strengths, since in most human diseases, phenotypes are complex and the relevant molecular mechanisms to target for therapy are not evident [79]. To overcome these problems and attempt to capture even complex phenotypes, these experimental screenings performed on cellular and animal models are very important for the possibility of being transferred to human treatment [80]. This work has demonstrated, for the first time, the ability of VGLCR to modulate the main important parameter implicated in glaucoma. These data support the hypothesis that in the future VGLCR could be the new strategy to slow down the degenerative process in human glaucoma, by activating the survival pathways, even in the initial phase of the disease. All these results have confirmed the active role of VGLCR in the repair of eye damage caused by glaucoma for the first time; VGLCR, therefore, appears to have greater beneficial effects if used preventively, becoming an important element in the fight against glaucoma.

## Figures and Tables

**Figure 1 foods-10-01885-f001:**
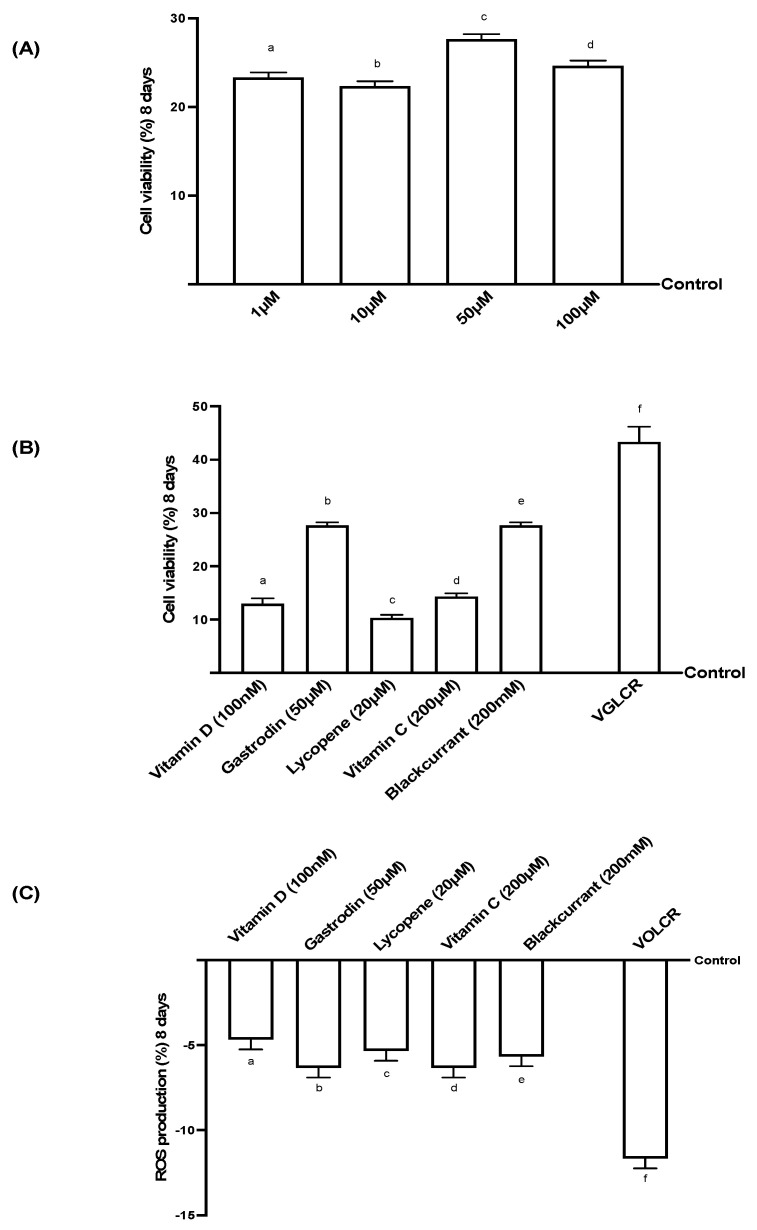
Cell viability and ROS production on RGC cells. In (**A**), dose-response study after gastrodin administration (1 µM–100 µM) on cell viability. Results are expressed as means ± SD (%) vs. control (0% line) of 4 independent experiments, each performed in triplicate. a,b,c and d is *p* < 0.05 vs. control; c is *p* < 0.05 vs. a,b and d. In (**B**), the effects of vitamin D, gastrodin, lycopene, vitamin C, blackcurrant, VGLCR = (V = vitamin D + G = gastrodin + L = lycopene + C = vitamin C + R = blackcurrant) on cell viability. In (**C**), the effects of vitamin D, gastrodin, lycopene, vitamin C, blackcurrant, VGLCR = (V = vitamin D + G= gastrodin + L= lycopene + C = vitamin C + R = blackcurrant) on ROS production. Both (**B**,**C**) results are expressed as means ± SD (%) vs. control (0% line) of 4 independent experiments, each performed in triplicate. a,b,c,d,e and f *p* < 0.05 vs. control; f is *p* < 0.05 vs. a,b,c,d and e.

**Figure 2 foods-10-01885-f002:**
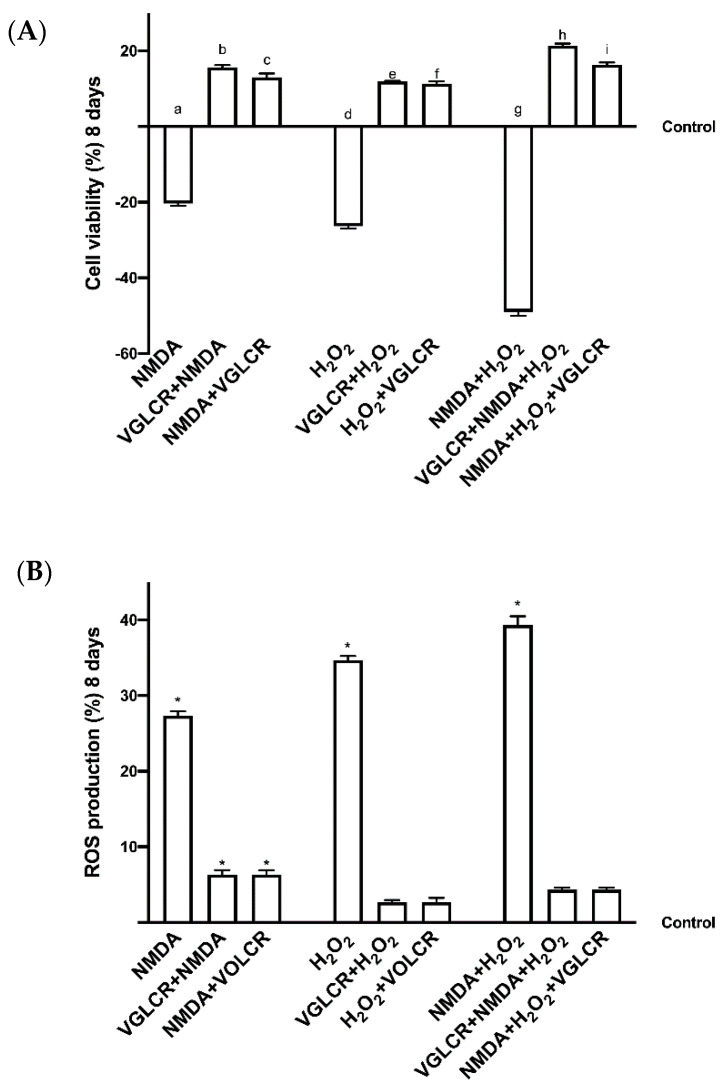
Cell viability and ROS production of RGC cells treated with NMDA and H_2_O_2_. In (**A**), cell viability is measured after NMDA and H_2_O_2_ alone and together, before and after VGLCR stimulation. Results are expressed as means ± SD (%) vs. control (0% line) of 4 independent experiments, each performed in triplicate. a–i *p* < 0.05 vs. control; b and c *p* < 0.05 vs. a; c *p* < 0.05 vs. b; e and f *p* < 0.05 vs. d; i and h *p* < 0.05 vs. g; I *p* < 0.05 vs. h. In (**B**), the same stimulation used on panel (**A**) to analyze ROS production. Results are expressed as means ± SD (%) vs. control (0% line) of 4 independent experiments each performed in triplicate. * *p* < 0.05 vs. control. Abbreviations are the same as used in Figure 1.

**Figure 3 foods-10-01885-f003:**
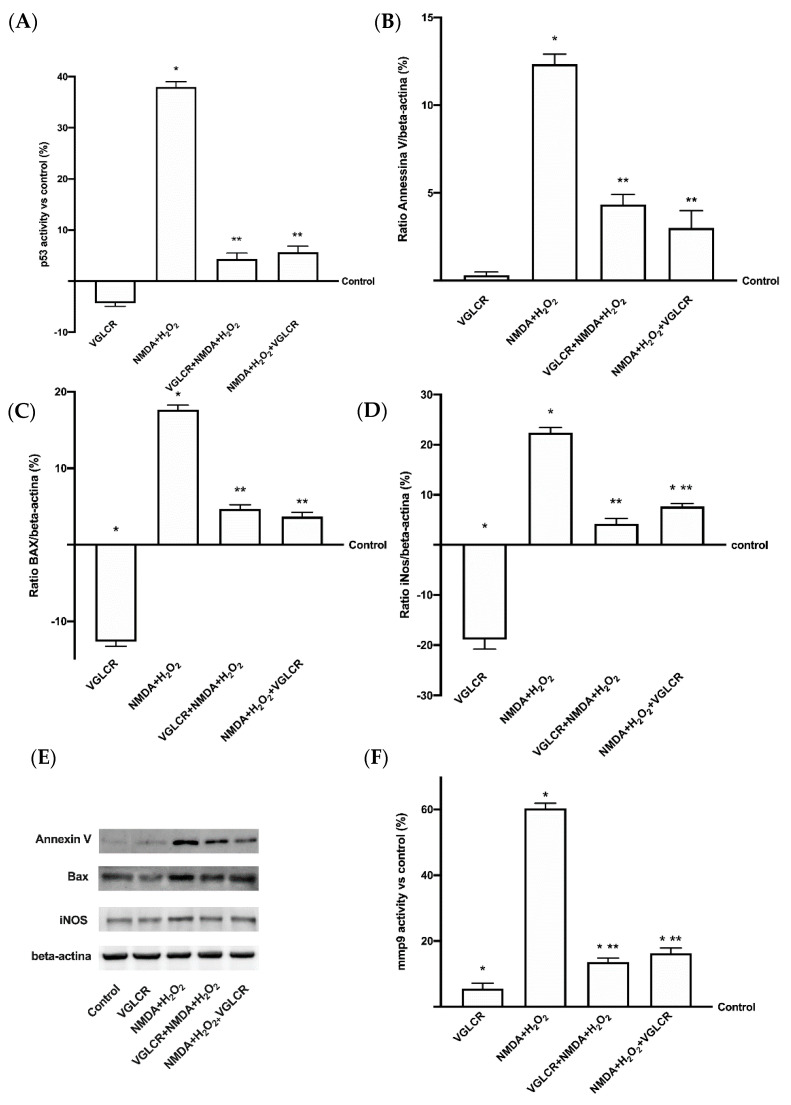
Western blot, densitometric analysis, and ELISA activity of main intracellular glaucoma markers. In (**A**) p53 activity measured by ELISA test; in (**B**) densitometric analysis of Annexin V; in (**C**), densitometric analysis of Bax; in (**D**) densitometric analysis of iNOS; in (**E**) Western blot of Annexin V, Bax, and iNOS; (F) MMP9 activity measured by ELISA test. All these markers were measured with NMDA+H_2_O_2_ before or after VGLCR stimulations. The results are expressed as means ± SD (%) vs. control (0% line) of 4 independent experiments, each performed in triplicate. Densitometric analysis are expressed as means ± SD (%) of 4 independent experiments, normalized and verified by β-actin detection. The images shown are an example of each protein from 4 independent experiments. * *p* < 0.05 vs. control; ** *p* < 0.05 vs. NMA+ H_2_O_2_. The abbreviations are the same as used in Figure 1.

**Figure 4 foods-10-01885-f004:**
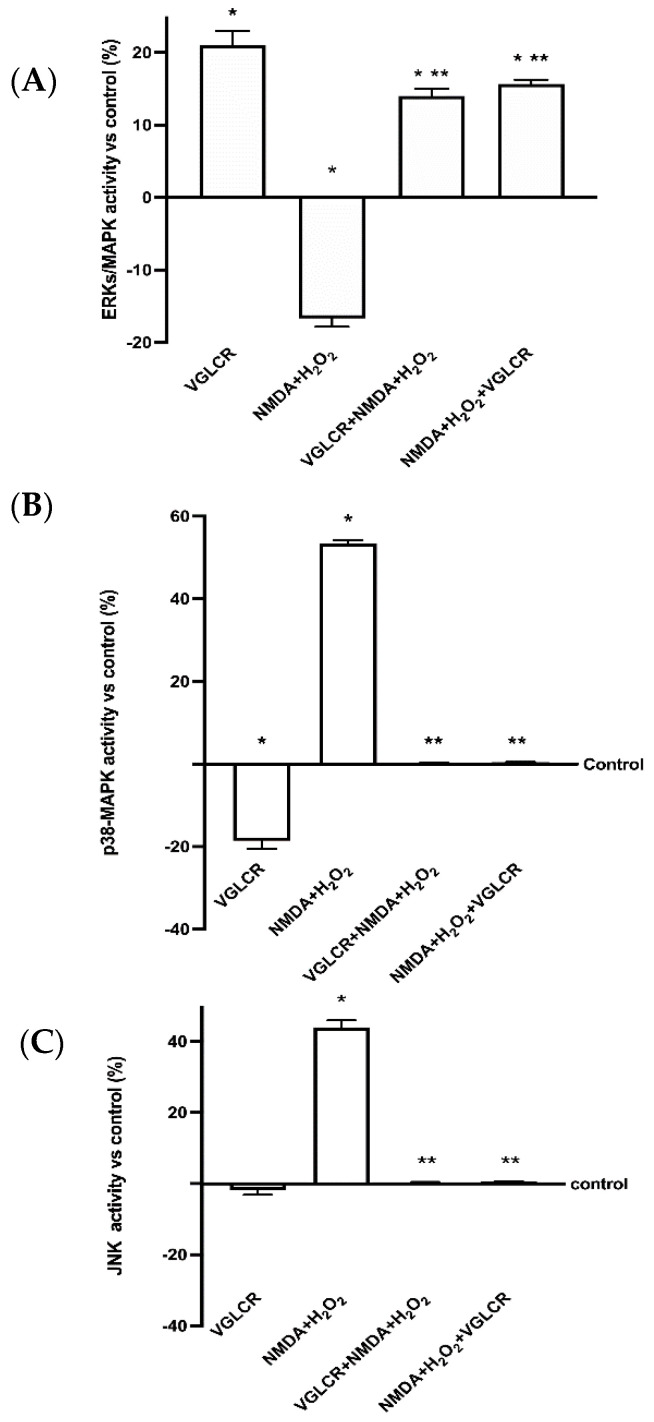
ELISA activity of ERKs (**A**), p38 (**B**), and JNK1/2 (**C**). All these markers were measured with NMDA and H_2_O_2_ added together, and before and after VGLCR stimulations. The results are expressed as means ±SD (%) vs. control (0% line) of 4 independent experiments, each performed in triplicate. * *p* < 0.05 vs. control; ** *p* < 0.05 vs. NMA + H_2_O_2_. The abbreviations are the same as used in Figure 1.

**Figure 5 foods-10-01885-f005:**
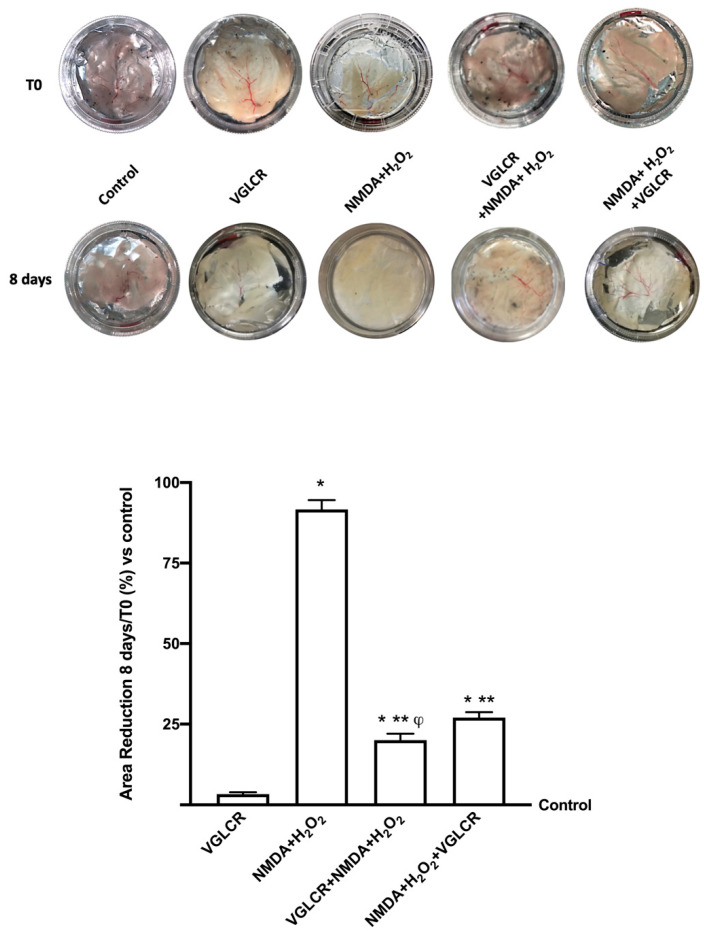
Morphological tissue and relative counting area on retinal tissues treated with NMDA and H_2_O_2_ added together, and before and after VGLCR stimulations. The results are expressed as a reduction area normalized on T0 and the control quantified by ImageJ software, expressed as means ±SD (%) of 4 independent experiments, reproduced in duplicates. * *p* < 0.05 vs. control; ** *p* < 0.05 vs. NMA + H_2_O_2_; ϕ *p* < 0.05 vs. NMA + H_2_O_2_ + VGLCR. The abbreviations are the same as used in Figure 1.

**Figure 6 foods-10-01885-f006:**
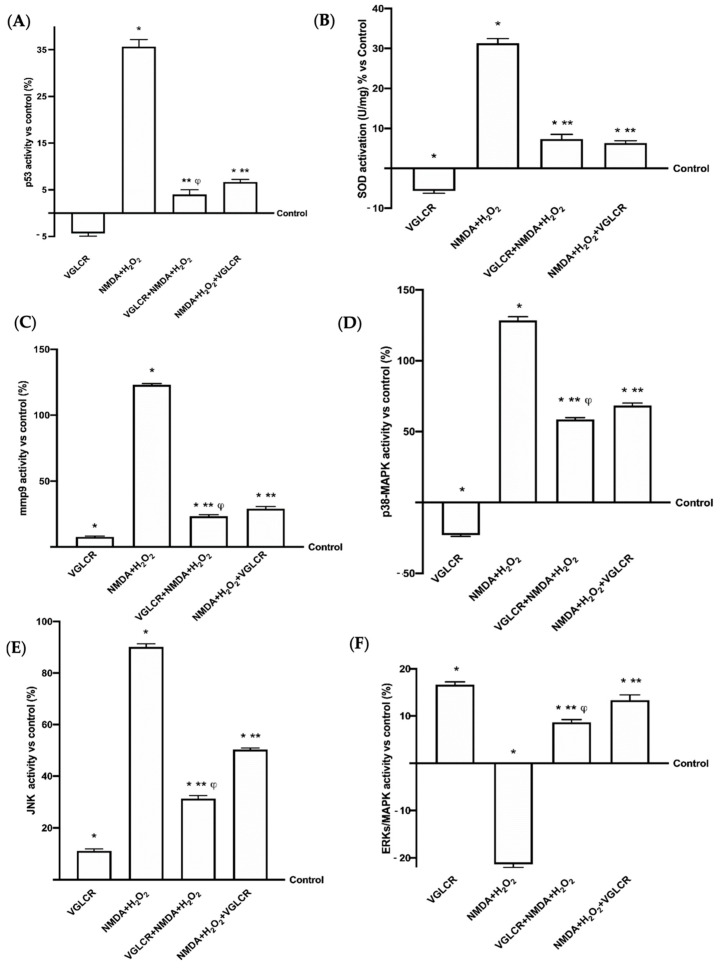
Activity assays of the main markers involved in glaucoma. In (**A**) p53, in (**B**) SOD, in (**C**) MMP9, in (**D**) p38/MAPK, in € Jnk1/2, in (**F**) ERKs/MAPK. Retinal tissues were treated with NMDA and H_2_O_2_ added together, and before and after VGLCR stimulations. The results are expressed as means ± SD (%) vs. control of 4 independent experiments, reproduced in duplicates. * *p* < 0.05 vs. control; ** *p* < 0.05 vs. NMA + H_2_O_2_; ϕ *p* < 0.05 vs. NMA + H_2_O_2_ + VGLCR. The abbreviations are the same as used in Figure 1.

**Figure 7 foods-10-01885-f007:**
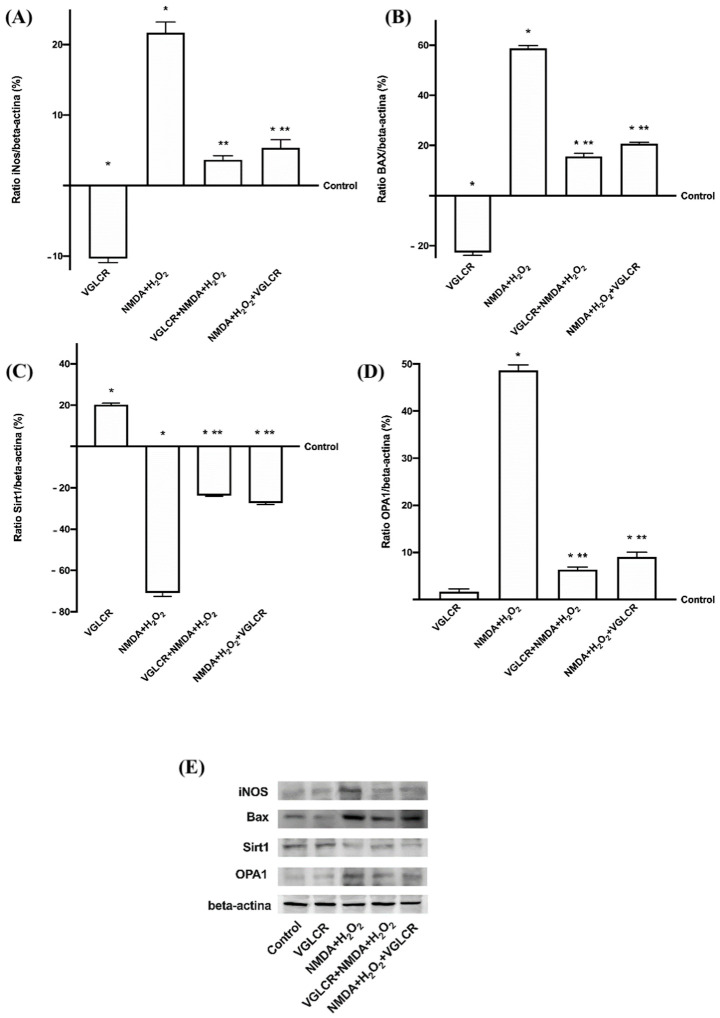
Western blot and densitometric analysis. In (**A**) iNOS, in (**B**) Bax, in (**C**) SIRT1, in (**D**) OPA1, and in (**E**) Western blot of iNOS, Bax, SIRT1, and OPA1. Retinal tissues were treated with NMDA and H_2_O_2_ added together, and before and after VGLCR stimulations. The images shown are an example of each protein obtained from four independent experiments. The results of 4 independent experiments performed in duplicates are expressed as means ± SD (%) vs. control, normalized and verified on β-actin detection. * *p* < 0.05 vs. control; ** *p* < 0.05 vs. NMA + H_2_O_2_; VGLCR + NMDA + H_2_O_2_ *p* < 0.05 vs. NMDA + H_2_O_2_ + VGLCR. The abbreviations are the same as used in Figure 1.

## Data Availability

Raw data are preferably deposited at Laboratory of Physiology (C. Molinari), ensuring appropriate measures so that raw data are retained in full forever under a secure system. The raw data are available under reasonable requested.

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
