# Peer review of "Effects of a New Combination of Natural Extracts on Glaucoma-Related Retinal Degeneration"

_foods, 2021, doi:10.3390/foods10081885_

Round 1

Reviewer 1 Report

It is interesting paper the translational study for treatment of glaucoma. Authors used FM-LipoMatrix for poorly water-soluble compound such as vitamin D3, gastrodin, lycopene and blackcurrant. They determined oxidative stress, neuroinflammation, apoptosis process in RGC culture and retinal explants. And they lead to the combination treatment was the most effective on NMDA-H2O2 induced RGC and retinal explants glaucoma model. The combination was VGLCR, instead of vitamin d and gastrodin. Therefor, the title should be changed to “Effect of VGLCR on glaucoma-related retinal degeneration” and the abstract should be rewrote following the results.

Minor changes

1) 2.4 Experimental protocol

Author should write clearly the treatment time difference between “NMDA+H2O2 and VGLCR combination” or “VGLCR combination and NMDA+H2O2”, respectively.

2) Author should reconcile figure legends and figure display.

Fig.3, line 412, (a)->(A), (b)->(B)

Fig.4. line 439 (a)->(A), (b)->(B), (c)->(C)

Fig.6, line 515 (b)->(B)

Fig.7 display disappear.

     Line 536 (a)->(A), (b)->(B), (c)->(C), (d)->(D)

     Line 537 (e)->(E)

3.Results

Line 286 (a)->(A), line 294 (b)->(B), line 306 (c )->(C ), line 326 (a)->(A), line 340 (b)->(B), Line 362 a->A, line 369 b,c,e->B,C, E, line 378 c, e->C, E, line 390 d, e->D, E, line 400 f->F, line 422 a->A, line 471 a->A, line 480 a,e->A, E, line 492 c,b,e->C,B, E, line 500 d,e->D, E, line 510 f->F, line 524 c,e->C,E, line 531 d,e->D, E

Author Response

Reviewer 1

It is interesting paper the translational study for treatment of glaucoma. Authors used FM-LipoMatrix for poorly water-soluble compound such as vitamin D3, gastrodin, lycopene and blackcurrant. They determined oxidative stress, neuroinflammation, apoptosis process in RGC culture and retinal explants. And they lead to the combination treatment was the most effective on NMDA-H2O2 induced RGC and retinal explants glaucoma model. The combination was VGLCR, instead of vitamin d and gastrodin.

Thank you for the comment we have made some changes according to your suggestions and we hope that now the manuscript is suitable for publication.

Therefore, the title should be changed to “Effect of VGLCR on glaucoma-related retinal degeneration” and the abstract should be rewrote following the results.

Thanks for the comment and we agree that the title needs to be changed. However, since the acronym VGLCR is not in common use, what do you think if we change the title to " Effect of a new combination of natural extracs on glaucoma-related retinal degeneration " ?

Minor changes

1) 2.4 Experimental protocol

Author should write clearly the treatment time difference between “NMDA+H2O2 and VGLCR combination” or “VGLCR combination and NMDA+H2O2”, respectively.

Thank you for the comment. We have added one phrase in experimental protocol section and changed the statement to improve the quality of the manuscript. All changes are reported in red.

2) Author should reconcile figure legends and figure display.

Fig.3, line 412, (a)->(A), (b)->(B)

Fig.4. line 439 (a)->(A), (b)->(B), (c)->(C)

Fig.6, line 515 (b)->(B)

Fig.7 display disappear.

     Line 536 (a)->(A), (b)->(B), (c)->(C), (d)->(D)

     Line 537 (e)->(E)

Thank you for the comment and we apologize for the inconvenience. We checked and changed all figures and legends as suggested. All changes are reported in red.

3.Results

Line 286 (a)->(A), line 294 (b)->(B), line 306 (c )->(C ), line 326 (a)->(A), line 340 (b)->(B), Line 362 a->A, line 369 b,c,e->B,C, E, line 378 c, e->C, E, line 390 d, e->D, E, line 400 f->F, line 422 a->A, line 471 a->A, line 480 a,e->A, E, line 492 c,b,e->C,B, E, line 500 d,e->D, E, line 510 f->F, line 524 c,e->C,E, line 531 d,e->D, E

Thank you for the comment and we apologize for the inconvenience. We checked and changed all results as suggested. All changes are reported in red.

Thank you for all usefully suggestions and we hope that now the manuscript will be acceptable for publication on “Foods” journal as open access form.

We added an author (N. Clemente) who helped us to perform further experiments to speed up the work and meet the deadline. All authors have approved the new version of manuscript for submission.
Thanking you for your attention,
Sincerely yours,

Reviewer 2 Report

In this manuscript, Molinari et al., research group developed a new oral formulation that were able to counteract the early changes connected to glaucomatous degeneration using the composition of gastrodin and vitamin D. This study looks interesting. However, there are few comments should be considered.

Comments:

  1. This paper strictly needs extensive revision for English which includes scientific terms in certain sentences which encourages the readers to understand clearly. There are several typo error throughout the manuscript.
  2. In Fig 1. when authors represent cell viability assays, the drugs should be screened for various time points to check its potency, authors are suggested to perform various time points for cell viability assay.
  3. Only one cell line were used for this study which is not sufficient to explain that this drug act as a promising drug candidate
  4. Did authors perform MMP-9 expression using gelatin zymography as this is a standard method for MMP-9 expression?
  5. For ROS production, authors should have used GSH as this is a universal ROS scavenger.
  6. Did authors perform all experiments independently in triplicates? Is there any computational simulation analysis was done for the to check the interaction mentioned in this study?
  7. Did authors perform any stress related pathway experiment other than ROS generation study, as this may be an important experiment to see whether any stress mechanism has an impact with the study? Authors are highly suggested to perform other stress related experiments.

Reviewer 3 Report

Dear Editor,

The manuscript by Molinari and Ruga et al. investigates the effectiveness of a combination based on vitamin D3 and gastrodin to counteract the course of retinal ganglion cells (RGC) degeneration.

The design of the study and the technical quality of the work look convincing and results can be of general interest. The manuscript is well-written and the authors used correct statistical approaches to analyse their findings. The methods are appropriate, described in good details (for most parts) and properly conducted. The claims are fully supported by the experimental data.

However, there is a number of major and minor points that would need to be addressed in order to improve the quality of this paper :

General:

-Materials: authors need to include the product numbers for all the kits and reagents. This is essential for the reproducibility of the data.

Major:

-Introduction lines 54-57. “Unfortunately, these methods have side effects that affect the quality of life [17]. In addition, there are topical pharmacological applications exerting an indirect neuroprotective action able to reduce IOP (such as α2-agonists, β-antagonists/blockers, prostaglandin analogs, carbonic anhydrase and cholinergic agents)”. This statement isn’t strictly true since recent studies have indicated that anti-edema therapies not only reduce the pressure but also improve motor and sensory functions. A breakthrough study by Kitchen et al Cell 2020 showed that targeting astrocytes using trifluoperazine following ischemia and hypoxia not only reduces edema but also stabilises the BBB/BSCB barriers and led to accelerated functional recovery compared with untreated animals. This role has been recently been confirmed by the work of Sylvain et al BBA 2021 which has demonstrated this viable therapeutic effect in pre-clinical stroke model. References to be included:

https://pubmed.ncbi.nlm.nih.gov/32413299/

https://pubmed.ncbi.nlm.nih.gov/33561476/

-Authors need to indicate the confluency at which they culture their cells and their doubling time since this can interfere with the interpretation of results if it hasn’t been kept consistent throughout the study. Moreover, authors need to indicate what passage range they have used for their cells in this study. This can be important to determine if the cells kept the phenotypic signature and how they have tested for this. A statement regarding their measures to investigate mycoplasma contamination needs to be mentioned as this can significantly interfere with the conclusion of this study.

-Discussion lines 565-567 “However, the use of in vitro models for the study of glaucoma opened the possibility to study the cellular and molecular mechanisms that may contribute to the clarification of disease onset and progression [62].”. Authors need to briefly discuss future directions which could include, but not limit to, the use of organoids and 3D cultures and organ-on-a-chip models especially those which are amenable for advanced imaging such as TEM and expansion microscopy since they enable real-time monitoring of neuroinflammatory processes. References to be included:

https://pubmed.ncbi.nlm.nih.gov/33117784/

https://pubmed.ncbi.nlm.nih.gov/32568450/

https://www.ncbi.nlm.nih.gov/pmc/articles/PMC6915910/

Minor:

-Conclusion: Retinal ganglion cells (RGC) degeneration and neurodegenerative diseases in general are yet incurable diseases. Authors need to point out to the recent advances in applying the use of high-throughput screening as has been nicely reviewed by Aldewachi et al 2021 and Varma et al as they can provide a novel insight that can support design of targeted therapies in future studies. References to be included:

https://pubmed.ncbi.nlm.nih.gov/33672148/

https://www.ncbi.nlm.nih.gov/pmc/articles/PMC2735222/

-Line 284: typo- performed.

Best.

Author Response

Reviewer 3

Dear Editor,

The manuscript by Molinari and Ruga et al. investigates the effectiveness of a combination based on vitamin D3 and gastrodin to counteract the course of retinal ganglion cells (RGC) degeneration. The design of the study and the technical quality of the work look convincing and results can be of general interest. The manuscript is well-written and the authors used correct statistical approaches to analyse their findings. The methods are appropriate, described in good details (for most parts) and properly conducted. The claims are fully supported by the experimental data.

Thanks for the comment. We have made some changes and hope that the manuscript is now sufficiently improved to be suitable for publication.

However, there is a number of major and minor points that would need to be addressed in order to improve the quality of this paper :

General:

-Materials: authors need to include the product numbers for all the kits and reagents. This is essential for the reproducibility of the data.

Thank you for the comment. In order to include all product number, we added in Material and Methods section (reported in red) the catalogue number (abbreviated as CN) after the Company name.

Major:

-Introduction lines 54-57. “Unfortunately, these methods have side effects that affect the quality of life [17]. In addition, there are topical pharmacological applications exerting an indirect neuroprotective action able to reduce IOP (such as α2-agonists, β-antagonists/blockers, prostaglandin analogs, carbonic anhydrase and cholinergic agents)”. This statement isn’t strictly true since recent studies have indicated that anti-edema therapies not only reduce the pressure but also improve motor and sensory functions. A breakthrough study by Kitchen et al Cell 2020 showed that targeting astrocytes using trifluoperazine following ischemia and hypoxia not only reduces edema but also stabilises the BBB/BSCB barriers and led to accelerated functional recovery compared with untreated animals. This role has been recently been confirmed by the work of Sylvain et al BBA 2021 which has demonstrated this viable therapeutic effect in pre-clinical stroke model. References to be included:

https://pubmed.ncbi.nlm.nih.gov/32413299/

https://pubmed.ncbi.nlm.nih.gov/33561476/

Thank you for the comment. We add a phrase after the “In addition, there are topical pharmacological applications exerting an indirect neuroprotective action able to reduce IOP (such as α2-agonists, β-antagonists/blockers, prostaglandin analogs, carbonic anhydrase and cholinergic agents) [18] to insert the references suggested as 19 [Kitchen P et al.] and 20 [Sylvain NJ at al.]. All changes are reported in red.

-Authors need to indicate the confluency at which they culture their cells and their doubling time since this can interfere with the interpretation of results if it hasn’t been kept consistent throughout the study. Moreover, authors need to indicate what passage range they have used for their cells in this study. This can be important to determine if the cells kept the phenotypic signature and how they have tested for this. A statement regarding their measures to investigate mycoplasma contamination needs to be mentioned as this can significantly interfere with the conclusion of this study.

Thanks for the comment. Cells obtained from retinal isolation take a long time to complete attachment to the surface. For this reason it is recommended not to change the soil for the first 72 hours. After 72 h, the attack is strong enough and the cells will not detach when the medium is vacuumed. In the following days the cells will expand while maintaining their pigmentation and after 2 weeks they will reach 80% confluence. Half of the medium was replenished every 2-3 days. [1]. The experiments were conducted immediately at 80% confluence in order to have a correct cell morphology and prevent contamination. This information has been added to the manuscript and shown in red. Indeed, primary cell cultures and early transition cultures have been reported to be less frequently contaminated with mycoplasma than continuous cell lines; primary cultures and first pass cultures of the order of 1 and 5%, respectively; cell lines in continuous culture in the range of 15-35%. For this reason we think that it is not necessary to check for contamination as indicated by Drexler and Uphoff [2].

  1. Kobayashi W, Onishi A, Tu HY, Takihara Y, Matsumura M, Tsujimoto K, Inatani M, Nakazawa T, Takahashi M. Culture Systems of Dissociated Mouse and Human Pluripotent Stem Cell-Derived Retinal Ganglion Cells Purified by Two-Step Immunopanning. Invest Ophthalmol Vis Sci. 2018 Feb 1;59(2):776-787. doi: 10.1167/iovs.17-22406
  2. Drexler HG, Uphoff CC. Mycoplasma contamination of cell cultures: Incidence, sources, effects, detection, elimination, prevention. Cytotechnology. 2002 Jul;39(2):75-90. doi: 10.1023/A:1022913015916

-Discussion lines 565-567 “However, the use of in vitro models for the study of glaucoma opened the possibility to study the cellular and molecular mechanisms that may contribute to the clarification of disease onset and progression [62].”. Authors need to briefly discuss future directions which could include, but not limit to, the use of organoids and 3D cultures and organ-on-a-chip models especially those which are amenable for advanced imaging such as TEM and expansion microscopy since they enable real-time monitoring of neuroinflammatory processes. References to be included:

https://pubmed.ncbi.nlm.nih.gov/33117784/

https://pubmed.ncbi.nlm.nih.gov/32568450/

https://www.ncbi.nlm.nih.gov/pmc/articles/PMC6915910/

Thanks for the comment. We agree with the suggestion on the importance of using organoids in the future. We have discussed this issue and added the suggested references. The sentences added in the Conclusion section are written in red.

Minor:

-Conclusion: Retinal ganglion cells (RGC) degeneration and neurodegenerative diseases in general are yet incurable diseases. Authors need to point out to the recent advances in applying the use of high-throughput screening as has been nicely reviewed by Aldewachi et al 2021 and Varma et al as they can provide a novel insight that can support design of targeted therapies in future studies. References to be included:

https://pubmed.ncbi.nlm.nih.gov/33672148/

https://www.ncbi.nlm.nih.gov/pmc/articles/PMC2735222/

Thank you for the comment. This point is very important, and we thank the reviewer for the references, which we have added in Conclusion section, written in red.

-Line 284: typo- performed.

Thank you for the comment and we apologize for mistake. We have carefully revised the manuscript to edit the typos (changes are reported in red)

Thank you for all usefully suggestions and we hope that now the manuscript will be acceptable for publication on “Foods” journal as open access form.

We added an author (N. Clemente) who helped us to perform further experiments to speed up the work and meet the deadline. All authors have approved the new version of manuscript for submission.
Thanking you for your attention,
Sincerely yours,

Round 2

Reviewer 3 Report

Dear Editor,

The authors have successfully addressed the majority of my comments and concerns in order to improve the quality of the manuscript.

I believe that the new section, improved methodology, and updated references, have contributed to enhancing the clarity of the manuscript, which I can now endorse for publication.

All the best!